# Mechanisms of Choice in X-Chromosome Inactivation

**DOI:** 10.3390/cells11030535

**Published:** 2022-02-03

**Authors:** Giulia Furlan, Rafael Galupa

**Affiliations:** 1Wellcome Trust/Cancer Research UK Gurdon Institute, University of Cambridge, Tennis Court Road, Cambridge CB2 1QN, UK; 2Department of Genetics, University of Cambridge, Downing Street, Cambridge CB2 3EH, UK; 3Developmental Biology Unit, European Molecular Biology Laboratory, 69117 Heidelberg, Germany

**Keywords:** X-chromosome inactivation, allelic choice, skewing, placentals, marsupials

## Abstract

Early in development, placental and marsupial mammals harbouring at least two X chromosomes per nucleus are faced with a choice that affects the rest of their lives: which of those X chromosomes to transcriptionally inactivate. This choice underlies phenotypical diversity in the composition of tissues and organs and in their response to the environment, and can determine whether an individual will be healthy or affected by an X-linked disease. Here, we review our current understanding of the process of choice during X-chromosome inactivation and its implications, focusing on the strategies evolved by different mammalian lineages and on the known and unknown molecular mechanisms and players involved.

## 1. X-Chromosome Inactivation: A Special Case of Dosage Compensation between the Sexes

In several taxonomic groups of animals, including nematodes, insects and mammals, the sex of a new individual is determined by sex chromosomes. The evolution of sex chromosomes has meant that the different sexes might end up with different dosages of sex-chromosome products; in humans, this would result in twice as much dose of ~1000 X-linked genes in XX individuals compared to XY individuals. While for some species these differences appear to be tolerable (reviewed in [1]), many others have evolved dosage compensation mechanisms to equalize sex-chromosome-linked gene expression between the sexes. Several strategies are known; for instance, in the fruit fly *Drosophila melanogaster*, hypertranscription of the X chromosome in XY individuals ensures an equal dose of X-linked products between XY and XX individuals. In the soil nematode *Caenorhabditis elegans*, dosage compensation happens in XX individuals (hermaphrodites) resulting in halved expression of both X-chromosomes, hence reducing global X-linked gene expression to the levels of the single X chromosome of XO individuals (males). In most mammals studied, dosage compensation also happens in XX individuals, but instead occurs via the transcriptional silencing of one of the X chromosomes, a process known as X-chromosome inactivation (XCI) (or “lyonisation”, after geneticist Mary Lyon, who first proposed such mechanism to occur [2]). Contrary to X-hypertranscription in XY flies or transcriptional repression of both X chromosomes in nematodes, XCI in XX mammals involves the differential treatment of two homologous chromosomes sharing the same nucleoplasm. While one X chromosome needs to remain transcriptionally active, the other must be (almost) completely shut down. In this review, we discuss our current understanding of how this choice is made across different mammalian taxa, and the molecular players underlying different strategies to achieve it.

## 2. Types of XCI Choice across Mammals: Predetermined or Rolling Dice

Different mammals have developed distinct strategies to accomplish X-linked dosage compensation. In extant prototherian mammals (monotremes, the egg-laying mammals), which possess multiple sex-chromosomes, chromosome-wide X-inactivation is absent, and dosage-compensation occurs in a locus- and tissue-specific manner [3]. In therian mammals, including the marsupial and placental clades, dosage compensation for X-linked gene products is achieved by nearly fully silencing one of the two X chromosomes in XX individuals. The need for this selective silencing brings about the problem of “choice”: how do these mammals choose which one of the two Xs to inactivate? While some species have solved this problem by always selecting the same X, for other species the process seems to be resolved rather randomly (Figure 1).

In marsupials (metatherians), the paternal X chromosome is exclusively (100%) chosen for inactivation [4], in a process known as “imprinted” X-chromosome inactivation (iXCI). The situation is more diverse in placental mammals (eutherians): iXCI occurs only in specific species and developmental stages and/or tissues, and the prevalent form is “random” X-chromosome inactivation (rXCI), observed in adult somatic tissues [5]. During rXCI, and in the absence of “skewing” influences, both the paternal X (Xp) and the maternal X (Xm) have roughly the same (~50%) chance of being inactivated.

In the murine preimplantation embryo, the most well-studied model for XCI research, a first wave of XCI following zygotic genome activation results in the exclusive inactivation of the Xp (Figure 1). At the late blastocyst stage, inactivation of the Xp is maintained in the extra-embryonic lineages, but reversed in the cells that will give rise to the embryo proper, which subsequently undergo random XCI upon implantation [6,7,8,9,10,11,12]. Imprinted XCI is also observed in the extra-embryonic cell lineages of rats [13] and cows [14,15]. In humans, early studies in trophoblast cells argued that the Xp is preferentially inactivated in this extra-embryonic tissue [16,17]; however, subsequent allele-specific analyses have concluded that XCI is in fact random in the placenta as well [18], with possibly only a slight bias towards the Xp [19]. Likewise, random XCI seems to be the only form of X inactivation in rabbits [20], pigs [21,22] and cynomolgus monkeys [23], both in embryonic and extra-embryonic cell types.

At the molecular level, imprinted and random XCI share some mechanistic features: both are regulated by a region on the X chromosome named the “X-inactivation centre”, *Xic* (though non-homologous between marsupials and placental mammals) and both are associated with the action of long non-coding RNAs (lncRNAs) that coat the X chromosome in *cis* and are proposed to direct gene silencing–for a recent review see [24]. In placental mammals, the lncRNA *Xist* is considered the critical trigger of XCI: this has been shown genetically in mice for both imprinted and random forms [25,26,27]. In marsupials *Xist* is not conserved, but a lncRNA with *Xist*-like properties, *Rsx*, has been recently identified in the gray short-tailed opossum: *Rsx* is expressed from and accumulates on the inactive X in XX cells and is able to silence genes in *cis* when transgenically inserted in mouse embryonic stem cells [28]. Accordingly, a recent methylome study in koalas found that the DNA methylation landscape upstream of *Rsx* showed a XX-specific pattern [29], consistent with another study in the opossum [30], altogether raising the possibility of *Rsx* being the functional analog of eutherian *Xist*.

## 3. Mechanisms of iXCI: Choosing to Inactivate the Xp

At the molecular level, imprinted XCI implies the existence of an epigenetic difference between Xp and Xm that would fully bias the choice towards the paternal chromosome. Rastan and colleagues, using uniparental embryos, showed that *Xist* expression is initially dictated solely by parental imprinting: paternal alleles are expressed and maternal alleles remain repressed, irrespective of X chromosome number [31]. The imprint could be in theory carried by the Xp, in a way that would promote XCI in *cis*, or by the Xm, in a way that would prevent XCI in *cis*. In marsupials, the molecular underpinnings of the imprint remain unknown, but a lot more has been investigated in the mouse, given its imprinted form of XCI during preimplantation development. Over the years, several hypotheses have been postulated regarding the nature of the imprint and the molecular mechanisms that lead to the inactivation of the Xp (Figure 2).

### 3.1. The First Proposals: A Paternal Imprint

Early studies postulated that the paternal X chromosome is intrinsically prone to inactivation, perhaps due to different levels of DNA methylation [32,33,34,35] or DNA condensation at the time of fertilization (reviewed in [36]). More recently, another hypothesis suggested that the Xp could retain an epigenetic memory acquired during its life cycle in the male: in both eutherians and metatherians, the imperfect pairing of the X chromosome with the Y chromosome during male gametogenesis results in the inactivation of both sex chromosomes during the pachytene stage of meiosis, in a process called meiotic sex chromosome inactivation (MSCI) [37,38]. According to this hypothesis, the Xp retained the epigenetic memory of MSCI silencing and entered fertilization in a pre-inactivated state [9,27,39]. However, it was later shown that the Xp, like the autosomes, is transcribed right after fertilization, at the time of zygotic genome activation, and only then is silencing initiated [27,40,41,42]. Moreover, Heard and colleagues showed that paternally inherited *Xist* transgenes that do not undergo MSCI are capable of inducing cis-inactivation [41], suggesting that MSCI is not necessary for iXCI in mice. Hence, the Xp is not inherited in a “pre-inactivated” state. This does not, however, answer the question of which parental X harbours the molecular imprint that leads to *Xist* expression from the Xp during early development.

The persistence of MSCI as a means of dosage compensation was postulated in metatherians and thought to explain how XCI was possible in the absence of a *Xist* homolog [43,44,45,46]. However, the observation that X-linked genes silenced by MSCI are reactivated after meiosis and subsequently re-inactivated in the female [47], refuted this hypothesis. As previously mentioned, an *Xist* analog was later found in marsupials, the lncRNA *Rsx* [28], but the molecular nature of the imprint in this clade remains unknown.

### 3.2. The Unexpected Outcome: A Non-Canonical Maternal Imprint

Imprinted XCI in mice has also been postulated to rely on a maternal imprint [31,48], later shown to be established in the female germ line and to prevent the Xm from being silenced during early embryogenesis [49]. In agreement with the “maternal imprint” hypothesis, studies on mouse embryos carrying extranumerary X chromosomes showed that they died early in development only when the extra chromosome was the maternal X [50,51,52], presumably due to failure of either Xm to inactivate (and ensuing lack of dosage compensation). Interestingly, this contrasts with the case of humans: XmXmY (Klinefelter syndrome) and XmXmXp (triple-X syndrome) individuals are viable, consistent with the lack of imprinted XCI in our species.

Importantly, the maternal imprint in mice is lost at the morula stage, during preimplantation development, as observed in uniparental embryos [31], and suggested by studies on diploid parthenogenetic embryos, which have two Xm chromosomes (no Xp)–in these embryos, inactivation of one of the Xm eventually occurs, both in the embryo proper and the extra-embryonic tissues [53,54].

Recently, a large-scale nuclear transfer study using donor cells from different stages of gametogenesis and embryogenesis showed that paternal X chromosomes always underwent inactivation during the time-window of imprinted XCI, and maternal X chromosomes unless derived from fully grown oocytes [55]. This confirmed that a maternal imprint is established late in oogenesis, and erased in embryonic but also extraembryonic lineages, where an imprinted XCI pattern persists. The loss of the imprint during preimplantation development suggests that other mechanisms might underlie iXCI in the extraembryonic lineages, which are being specified around the time the imprint is lost.

Canonical genomic imprinting relies on allele-specific DNA methylation (see [56] for a recent review), and such mechanism was initially proposed to underlie iXCI as well [33,34,35]. Contrary to these early hypotheses, however, later studies with DNA methyltransferase maternal knockout embryos ruled out DNA methylation as having a role in setting the imprint [57]. Recently, Yi Zhang’s lab identified the polycomb-repressive complex 2 (PRC2)-dependent mark, H3K27me3, as a DNA methylation-independent mechanism underlying the imprinted patterns of several loci across the genome [58], including the *Xist* locus [59]. In particular, a large H3K27me3 domain was found to coat the *Xist* locus during oocyte growth and to persist through preimplantation development, with loss of maternal H3K27me3 resulting in ectopic *Xist* expression from the Xm and maternal XCI in preimplantation embryos [59]. Interestingly, this H3K27me3 domain at *Xist* and at other maternally-imprinted loci coincides with maternally-specific topological domains in early embryos as detected by Hi-C, a chromosome conformation capture technique [60]. Of note, another repressive histone mark, H3K9me3, has been shown to be enriched at the *Xist* locus in early preimplantation embryos [61] but its role remains contested: while overexpression of *Kdm4b*, a H3K9me3 demethylase, partially derepresses *Xist* on the Xm in parthenogenetic embryos [61], injection of *Kdm4b* mRNA into in vitro fertilization-derived biparental embryos does not result in *Xist* derepression on the Xm in either XY or XX embryos [59].

While the chemical nature of the imprint has finally been discovered, it is still not clear which sequences are critical to carry the imprint and/or for the imprint mark to be laid. The maternal H3K27me3 domain spans ~450 kb, including the *Xist* locus and its positive regulators *Jpx* and *Ftx* [62,63]. The critical sequences are most likely contained within a subregion of that domain, given that a 210 kb transgene that contains *Xist*, *Jpx*, and part of *Ftx,* as well as *Xist*’s negative antisense regulator *Tsix* [64], can recapitulate the imprinted expression of *Xist* in early mouse development [41]. *Tsix*–which is not covered by the H3K27me3 domain–was initially implicated as the imprinted locus, given that disruption of *Tsix* on the Xm leads to *Xist* upregulation in *cis* and results in post-implantation death in females due to impaired development of extraembryonic lineages [65,66]. However, at the 4-cell stage, when *Xist* starts to be upregulated on the Xp, *Tsix* is still silent on the Xm [66], suggesting that the maternal imprint is independent of *Tsix* transcription–which seems to be important at later stages, in the extraembryonic lineages, to *maintain* the imprinted pattern of XCI.

Regarding *Ftx* and *Jpx,* deletions of either alone show that they are dispensable for iXCI in preimplantation embryos [60,67]. However, maternal transmission of a ~115 kb deletion that spans the *Jpx* locus, part of the *Ftx* locus, and the intergenic region in between them led to compromised viability of XX embryos and no XY pups being born [60], a pattern compatible with loss of the maternal imprint. It remains to be investigated how exactly such sequences contribute to the imprinted expression of *Xist*, if important for the establishment of the imprint (H3K27me3 deposition) during oogenesis, or for its maintenance, and whether other regions might be necessary as well to carry the imprint (e.g., the *Xist* promoter).

In marsupials, whether the *Rsx* locus carries an imprint on either the Xp or the Xm or both is not fully known, but a non-coding RNA antisense to *Rsx* and *Xsr,* has been recently discovered and shown to exhibit a *Tsix*-like behavior [68]. *Xsr* is expressed from the Xm in the early embryo, but not in adults, and is thought to prevent *Rsx* expression in *cis* [68]. Early *Xsr* expression from the Xm could, in marsupials, play an imprint role equivalent to *Tsix* during the maintenance phase of XCI in murine extraembryonic tissues.

Overall, a rather complex picture emerges, which favors the existence of a maternal imprint, at least in mice: during oogenesis and in the pre-implantation embryo, chromatin condensation and the deposition of repressive histone marks at the *Xist* promoter are associated with preventing *Xist* expression on the Xm. Later, in the extra-embryonic tissues of the post-implantation embryo, *Tsix* expression from the Xm prevents its silencing in *cis* by repressing *Xist* expression.

It is still possible, however, that the imprint is not carried by the Xm exclusively, but that a combination of maternal and paternal signals is needed to ensure monoallelic *Xist* upregulation on the Xp. For instance, chromatin structure at the *Xist* locus differs between the paternal and the maternal chromosomes at the time of fertilization: not only the maternal X carries repressive chromatin marks, but the paternal X undergoes chromatin decondensation when the sperm-derived genome of the paternal pronucleus is subjected to global replacement of protamines with histones [69]. Hence, it is possible that some loci, including *Xist*, are poised to be preferentially transcribed over their maternal counterparts in the zygote. Moreover, since silencing of repeats (LINEs, SINEs) is proposed to precede genic silencing in iXCI [27], it is also possible that the Xp is treated differently according to its epigenomic content, with repeat-rich regions being inherited in a pre-inactivated state through the paternal germline following MSCI [9,70] and gene-rich regions being inactivated *de novo* after ZGA. Interestingly, this could also be true for marsupials, where *cot-1* repeats are silenced during the final stages of spermatogenesis and are possibly inherited in a pre-inactivated state [47,71].

### 3.3. Evolutionary Considerations about iXCI

Imprinted XCI was initially hypothesized to represent the ancestral form of X inactivation in therian mammals, potentially present in the common ancestor between marsupials and placentals, which diverged 180 million years ago [39,72]. Imprinted XCI was initially thought to be partially conserved in mice, which also evolved random XCI, but lost in hominids, who evolved random XCI exclusively–reviewed in [73]. However, as it becomes clearer that the molecular players involved in marsupial iXCI and murine iXCI are different and not homologous, it seems more plausible that iXCI evolved independently (and convergently) in the phylogenetic lineages of mammals with pouches and mammals with placentas; the common ancestor had perhaps no X-linked dosage compensation mechanism, like monotremes.

The absence of imprinted *Xist* expression in non-rodent eutherian species, which in general have a later onset of XCI in early development, has been suggested to be linked to the fact that embryos of those species undergo several rounds of DNA replication before *Xist* starts to be expressed; this way, parent-specific chromatin structure differences would possibly be erased, resulting in the two *Xist* alleles being epigenetically identical [74,75]. A more recent model posits that an imprint that instructs strictly monoallelic XCI must exist in species with early XCI initiation (such as marsupials and mice) to prevent the possibility of inactivating both X chromosomes [68], which, nevertheless, happens during random XCI (discussed below). In early development the consequences of having two silent Xs (such as cell death) would be very detrimental to the developing embryo, contrary to later development, when more cells are present and the embryo could potentially afford to lose some [68]. Also, during random XCI, in post-implantation development, inactivation of both Xs can be reversed [76]. During iXCI such reversion would probably be difficult to trigger, given the fast paced kinetics of preimplantation development.

Albeit intriguing, these hypotheses do not explain why, in both marsupials and mice, it is always the paternal X but never the maternal X chosen for inactivation. A proposal by Heard and colleagues states that a maternal imprint is needed “to prevent the early activity of X-linked paternal genes involved in placental growth, as proposed in the parental conflict theory, particularly in rodents where zygotic gene activation occurs very early on in development” [41]. Interestingly, however, life is possible even if it is the Xp that remains active: XX mice with a paternally-inherited *Xist* deletion die during embryogenesis due to XCI failure in the extra-embryonic tissues [49,77], but this lethality can be rescued by deleting *Tsix* on the Xm, which then undergoes inactivation [66]. While this indicates that inactivating the Xm instead (and keeping the Xp active) can sustain life, it also suggests that the strength of the imprint is such that it is very difficult to override it. It is still unknown whether there would be decreased fitness due to inactivating the Xm in XX individuals.

On the other hand, had the opposite pattern of iXCI evolved in XX embryos–always inactivating the Xm–then mechanisms would have had to evolve in XY embryos to make sure that their Xm would always remain active during development. It is thus possible that a strong maternal imprint preventing early inactivation of the Xm evolved to limit the risk of XY offspring inheriting a pre-inactivated Xm, a condition that would be lethal during early development. 

## 4. Mechanisms of rXCI in Mouse: A Race for Inactivation

In the context of iXCI, it is intuitive to think about choice: given its parental origin, each chromosome has its fate predetermined. When talking about rXCI, we usually pose the question of choice in similar terms: how is one of the X chromosomes chosen at random to upregulate *Xist* thus becoming the inactive X chromosome? However, this is misleading. Contrary to what happens during iXCI, in which choice *precedes Xist* expression and gene silencing, choice during rXCI is only “set” *after Xist* is upregulated and starts inducing (some) gene silencing in *cis*, which then prevents the other chromosome from doing the same (Figure 3). Thus, it is the beginning of the XCI process itself that determines choice–choice is not set from the start. Accordingly, we now know that both X chromosomes can start upregulating *Xist*: this has been observed both in the early mouse embryo and in cultured mouse embryonic stem cells undergoing differentiation [76]. This situation can be reversed [76], otherwise cells would die, and thus, the process resets and each chromosome has another opportunity to become the inactive X. Biallelic *Xist* expression is also observed in rabbits, monkeys and human embryos [20,23,78,79], but much less is known in these species about the regulation of *Xist* expression and the onset of XCI. These observations serve to illustrate how, preceding the stages in which random XCI is set, both chromosomes presumably have the potential to become the inactive X. Alternatively, one could imagine a scenario similar to what happens during iXCI: prior to the initiation of rXCI, one of the chromosomes would be “marked” at random to become the chosen one for inactivation. This chromosome would then upregulate *Xist*, which would lead to transcriptional silencing in *cis*. However, no such mark has been discovered–apart from the upregulation of *Xist* itself, which could be an indicator of which chromosome would be the inactive X (though not in 100% of the cases, as discussed above). This and other alternative scenarios have been thoroughly and elegantly reviewed in Mutzel and Schulz’s “systems biology perspective” on rXCI [80]. The model that reflects our current understanding of the process is that the “random” choice we observe in the tissues of XX mammals is a final result of which chromosome managed to start inactivating first while efficiently preventing the other chromosome from starting its own inactivation. The Schulz lab has recently established a theoretical framework (accompanied by experimental demonstrations) by which to think about the problem of random XCI [76]; this framework explains not only the process of choice, but also the process of counting, as well as observations of XCI patterns from cells with different numbers of X chromosomes and across species.

In the absence of a choice mechanism *before* the upregulation of *Xist* during random XCI, the question of choice–which X chromosome ultimately becomes the inactive X–morphs into two separate questions: what influences the upregulation of *Xist* from one chromosome or the other, and what mechanisms prevent one chromosome from upregulating *Xist* when the other one started doing so. 

### 4.1. Influencing Choice by Influencing Xist Upregulation

What do we know about how *Xist* might become upregulated asymmetrically? Both *cis* and *trans* mechanisms could be at play; *cis*-mechanisms by acting independently on each chromosome, and *trans*-mechanisms by underlying local (chromosomal) fluctuations (e.g., *trans*-factors concentrations) could affect chromosomes differentially. Certain *trans*-acting factors have been proposed to bind asymmetrically to the *Xist* promoter: YY1, RIF1 and KAP1. At the onset of XCI, RIF1 is associated with the X chromosome upregulating *Xist* and is critical for upregulation [81]. A very similar role has been proposed for YY1 [82], and KAP1 binds to the allele with no *Xist* expression, due to the absence of RIF1 [81]. The asymmetric binding of RIF1 (and KAP1) seems to be downstream of fluctuations of antisense transcription across the *Xist* locus [81], while YY1 binding is due to differential DNA methylation of *Xist* alleles at the onset of XCI [82]. This has also been reported downstream of antisense transcription across the *Xist* locus [83,84,85]. Together, these findings suggest that the asymmetric binding of these *trans*-factors serve as “bookmarks” for the future active and inactive X (though most likely not irreversibly), reinforcing asymmetries derived from *cis*-acting mechanisms.

Several loci are known to influence the regulation of *Xist* in *cis*, however, this is mainly based on knockout studies–i.e., we understand the consequences for XCI choice of the *absence* of such loci, but how they potentially regulate *Xist* asymmetrically when their two copies are present remains speculative. One such *cis*-acting loci is *Xist*’s antisense transcription unit, *Tsix*. Heterozygous deletions of its promoter region lead to nonrandom XCI patterns such that the mutant X chromosome is always the inactive X [64,86]. Thus, it seems that the absence of *Tsix* substantially increases the likelihood of *Xist* upregulation from the same chromosome, in such a way that it is always the mutant chromosome that upregulates *Xist* and becomes the future Xi–the other allele never has a chance to do it. One would expect *Tsix*-heterozygous cells to be quicker than wild type cells at upregulating *Xist* and initiating XCI; faster kinetics during differentiation have indeed been observed [87,88]. Additionally, one can conclude that *Tsix* normally represses *Xist* expression in *cis*, and further studies demonstrated that it is *Tsix* transcription (and not its RNA products) that is important for negative regulation [83,89,90]. The exact mechanisms seem to involve chromatin changes at the *Xist* promoter when *Tsix* is transcribed across it [83,84,91,92,93,94] and potentially transcription interference via polymerase clashes [76]. What are the implications for the initiation of XCI when both *Tsix* alleles are functional? Given that *Tsix* transcription seems to affect the likelihood of *Xist* expression in *cis*, asymmetries in *Tsix* transcription between the two X chromosomes could therefore generate asymmetries in *Xist* upregulation and therefore ultimately influence which X becomes inactive. However, we do not knowwhether, for instance, rates of *Tsix* transcription could modulate the probability of *Xist* to be upregulated–and if such a relationship would be linear or non-linear. Overexpression of *Tsix* from one allele exclusively has the expected effect of that chromosome remaining active [91]. However, no experiments have been done in which *Tsix* transcription is modulated and *Xist* expression followed at the single-cell, single-chromosome level. Importantly, while *Tsix* function is critical in mice, the locus is not well conserved across placental mammals; in humans, the locus is predicted to be present but its structure is different (it does not overlap with the entire *Xist* locus and promoter), raising questions of whether *TSIX* is at all functional [95,96,97].

While manipulations of *Tsix* seem to have an all-or-nothing effect regarding choice, other loci in the vicinity of *Xist* have milder but significant effects on choice patterns. Heterozygous deletions of *Xite* (an enhancer-like locus upstream of *Tsix*) or of the promoter of *Linx* (another lncRNA locus within the *X-inactivation centre*) lead to skewed patterns of choice, with the mutant chromosome being more often the inactive X [98,99]. These loci thus seem to be negative *cis*-regulators of *Xist*; *Xite* is thought to act via influencing *Tsix* transcription and the promoter of *Linx* acts on *Xist* independently of *Tsix* and presumably via a “silencer” (a “negative” enhancer) type of mechanism (and not via *Linx* transcription or transcripts). In each case, asymmetries in *Xite* or *Linx* function between the chromosomes could lead to asymmetries in terms of *Xist* upregulation during development and differentiation, though presumably with less influence than *Tsix*, as judged by the results of the knockout studies. Like *Tsix*, *Xite* is poorly conserved across placental mammals [97], while the promoter of *Linx* is well conserved, suggesting a more ancestral role as a *cis*-negative regulator [99]. Another mouse-specific locus known to influence choice is the *Xce*, which we discuss in a section below.

Interestingly, the three negative regulators of *Xist*–*Tsix, Xite* and *Linx*–all reside within the same topologically associating domain (TAD). TADs partition the mammalian genome in regions that include genes and *cis*-regulatory elements showing functional similarities, and such partitioning seems to be critical, at least for some developmentally-regulated loci–for a recent review, please see [100]. The *Xist*/*Tsix* unit–in mice and humans–lies at the boundary between two TADs [97,101,102], with each promoter associated with opposite, neighbouring TADs–as mentioned, the *Tsix* promoter lies within the same TAD as *Xite* and *Linx*, and the *Xist* promoter shares a TAD with loci such as *Jpx*, *Ftx, Xert* and *Rnf12*, which all have shown to be positive regulators of *Xist.* Genetic manipulations that invert the *Xist*/*Tsix* locus, changing the TADs in which the promoters are placed, lead to the misregulation of both *Xist* and *Tsix* and effects on XCI [103], highlighting how the *cis*-regulatory landscape of each promoter is critical for appropriate initiation of XCI.

Among the positive regulators of *Xist* within the same TAD as the *Xist* promoter, three lncRNA loci have been reported to influence *Xist* expression in *cis* and could therefore affect choice–*Ftx* and *Xert* via their transcription [62,104], and *Jpx* transcriptionally or post-transcriptionally [63,105]. *Jpx* has also been reported as a *trans*-acting regulator of *Xist* via its lncRNA [106]. A recent study has identified a series of proximal and distal enhancers that are also required for activation of *Xist* upregulation in *cis* [104]. Elements within the *Xist* locus itself have also been reported to contribute to *Xist* upregulation–e.g., [107], including an antisense transcript, *XistAR* [108]. 

In conclusion, loci that influence the upregulation of *Xist* can ultimately influence choice: the X chromosome that upregulates *Xist* first is more likely to become the Xi. Importantly, a *cis*-acting positive feedback mechanism has been predicted to be essential for establishing *Xist* monoallelic regulation during the initiation of XCI, and therefore critical for choice [76,80]. This *cis*-acting positive feedback mechanism reinforces *Xist* expression, either via mutual repression (*Xist* expression silences one of its *cis*-repressors) or mutual activation (*Xist* expression promotes expression of its *cis*-activators). Mutual repression is known to exist between *Xist* and *Tsix*: silencing of *Tsix* expression by *Xist* RNA alleviates *Tsix-*mediated repression of *Xist* expression. Recently, the protein SPEN, shown to be essential for gene silencing during XCI [109], has been implicated in this feedback mechanism: SPEN (recruited by *Xist* RNA) is required to silence the *Tsix* promoter, which in turn is required for consistent *Xist* upregulation [110].

### 4.2. Influencing Choice by Preventing Xist Upregulation from the Second Chromosome

Choice during random XCI does not depend exclusively on which X upregulates *Xist* first. This must be sustained on that chromosome, and the other X has to be prevented from upregulating *Xist* as well, only then is choice really established. How does this work at the molecular level? X-linked factors need to be present in a double dose to activate *Xist*–this model explains not only the process of choice but also why XCI happens in XX individuals but not in XY individuals, which never have a double dose of such X-linked factors. When one of the chromosomes upregulates *Xist*, its RNA induces silencing of genes in *cis*, including those coding for the X-linked activators, and their dose in the cell is reduced to half, which is not enough for the other chromosome to be able to upregulate *Xist* efficiently [111,112]. Hence, the fate of each chromosome becomes locked, as one becomes the Xi, and the other remains active—“choice” is thus determined, but only then. In cells in which both chromosomes upregulate *Xist*, the same model predicts that *cis*-silencing leads to a quick drop in the levels of X-linked activators, needed to sustain *Xist* upregulation. Therefore, *Xist* expression from both chromosomes aborts and the process starts again [76,112].

One such dosage-dependent X-linked activators is encoded by the *Rnf12/Rlim* locus: RNF12 is a ubiquitin ligase that targets for degradation of the pluripotency factor REX1, a repressor of *Xist* that binds to its promoter in undifferentiated cells [113,114,115]. In XY murine embryonic stem cells, extra copies of *Rnf12* can activate *Xist* expression, while in XX murine embryonic stem cells, knockout of *Rnf12* seems to abolish XCI during differentiation [113], although these latter results have been contested [116]. In vivo, *Rnf12* knockout is reported to prevent imprinted XCI but not the random form [116,117]; however, the inducible system used for knocking out *Rnf12* and study rXCI might not be the most suitable in terms of developmental timings [118], therefore, *Rnf12’*s role in rXCI remains open. We know, nevertheless, that there is more to the dosage-dependent regulation of XCI than *Rnf12*. For instance, in XX murine embryonic stem cells with a single copy of Rnf12 (heterozygous deletion), XCI is not abolished, but only delayed [113], suggesting that there are other factors capable of activating XCI in a dosage-dependent manner [80]. The other known X-linked activators of *Xist* and XCI (such as *Jpx* and *Ftx*) cannot account for the feedback loop necessary to prevent the upregulation of *Xist* from one chromosome when the other has already started it [80], therefore additional factors remain to be identified.

## 5. Choice in Human rXCI: Biallelic Dampening or Direct Monoallelic Inactivation?

X chromosome activity has specific dynamics in humans: like in the mouse, both Xs are transiently active in the inner cell mass of the blastocyst of the preimplantation XX embryo [20,78,119], but, surprisingly, at this developmental stage, *XIST* is expressed from both X chromosomes [20,78,79]. Indeed, the *XIST* RNA accumulates in *cis* forming a (typical) “cloud” as observed by RNA FISH [20,78], albeit without triggering H3K27me3 enrichment [20]. Curiously, the X chromosome in XY embryos also expresses *XIST* and shows cis-accumulation [20]. Together, these observations were taken to strongly suggest that *XIST* expression is uncoupled from XCI in human preimplantation development. A similar picture has emerged for another primate, the cynomolgus monkey, in terms of *XIST* expression in XX and XY embryos, as reported very recently in an unprecedented study characterizing XCI dynamics during development in a non-rodent species [23]. Choice thus seems to be preceded by *XIST* expression, like in the mouse, but in both human and cynomolgus monkeys it seems to be more about from which X to repress *XIST* expression.

Two models have been proposed to describe the initiation of dosage compensation in the human preimplantation embryo: X-dampening and direct X-inactivation–and similar considerations could be drawn for the cynomolgus monkey, see [23,120]. According to the dampening model, progressive increase in biallelic *XIST* expression results in a gradual biallelic downregulation of X-linked genes from morula to blastocyst [78], in a form of dosage compensation reminiscent of the strategy adopted by *C. elegans* hermaphrodites and comparable to an “absence of choice”. How this Xd/Xd state would later transition into an Xa/Xi state, which is observed in human somatic tissues, triggering a choice between two equal Xd chromosomes for one of them to become the Xi, unknown. X chromosome dampening has also been described in XX human pluripotent stem cells (hPSCs, the *ex-vivo* model of choice to investigate human XCI dynamics) during the conversion from primed to naïve pluripotency state: using bulk RNA-seq datasets, the authors have shown that initial reactivation of the inactive X chromosome from primed to early naïve state was subsequently followed by X-dampening in late naïve cells [121,122].

The dampening model remains controversial and has been contested both in preimplantation embryos and hPSCs–see [123,124,125] for reviews. While reanalyzing the same transcriptome dataset published in Petropoulos et al. (2016) with more stringent conditions, De Mello et al. observed a decrease in the proportion of biallelically expressed X-linked genes, which is consistent with XCI, and a constant level of their median expression hence refuting the hypothesis of X-dampening and suggesting that human initiation of dosage compensation rather occurs through direct X-inactivation [18]. In agreement with the direct X-inactivation model, Mandal et al. observed partial X-reactivation of the inactive X-chromosome rather than Xd/Xd dampening in hPSCs reverting from primed to naïve state [126], when re-analyzing published single-cell RNA-seq datasets [122]. Although the dynamics of human dosage compensation initiation are debated, an additional lncRNA, *XACT*, has been proposed to underlie choice [79]; interestingly, *XACT* accumulates in *cis* on both (active) X chromosomes in an “antagonistic” manner to *XIST*, i.e., regions of the X not covered by *XIST* are covered by *XACT*, as if they repel each other [79,127].

## 6. Preferences in Choice: Random XCI Patterns Are Often Skewed

In theory, both X chromosomes during random XCI have the same probability of being inactivated–this would of course be the case if both chromosomes were genetically identical (which does not occur in wild populations, including human ones, but happens in inbred strains in laboratories) or if, of their genetic differences, none would influence the mechanisms of choice. However, deviations from 50:50 in patterns of inactivation are often observed. This skewing can occur as (i) a result of stochasticity, (ii) due to non-random choice at the onset of XCI, known as ‘primary’ choice, or (iii) as a result of selection for or against cells carrying one specific active or inactive X chromosome, known as ‘secondary’ choice [128,129,130,131]. Random XCI skewing due to stochastic events implies that, in the absence of genetic differences at the *Xic* or mutations in any other part of the X chromosomes that significantly affect the mechanisms of choice, more than half of the cells in an embryo (or adult) end up with the same inactive X. This phenomenon has been observed in inbred mice that carry different parent-of-origin fluorescent tags on either of their chromosomes but are otherwise genetically identical [132]. These mice show a high degree of XCI mosaicism between littermate siblings and even across tissues in the same individual, with sometimes as much as 90% of the cells of a tissue carrying the same inactive X, based only (presumably) on “stochastic choice” [132].

In humans, several studies have investigated the prevalence of skewed XCI in “phenotypically unaffected” XX individuals and reported widely different results, with the percentages of individuals showing skewed XCI ranging from less than 10% to over 50% [133,134,135,136,137,138]; these variations could depend on the degree of skewing considered, the analysis method, the type of tissue analyzed, and/or the age of the persons. For instance, a study among one thousand XX human individuals found skewed XCI in a large proportion of phenotypically healthy individuals, with ~15% of adults exhibiting skewing greater than 80:20 in peripheral blood lymphocytes [135]. Whichever the degree of skewing, such stochastic-related imbalances reflect the ratio of “founder” cells to adult cells in specific tissues and organs, as well as the timing and extent of cell migration during development [132]; importantly, XCI is initiated at a time when the number of cells in the embryo is limited, hence achieving a perfect 50–50% inactivation ratio is not the most statistically likely event, already at the moment of XCI initiation [138]. Yet, given the high heterozygosity in the human population, another possibility is that the observed random skewing is in fact primary skewing, with individual variations (e.g., SNPs) at X-linked loci potentially leading to skewed XCI due to a preferential choice of one X chromosome over the other at the time of XCI initiation (analogous to *Xce* alleles in mice, discussed below). 

In primary skewing, potentially any variant in genes involved in the XCI process itself (usually genes within the *Xic*) could influence choice by having an impact on the upregulation of *Xist* and/or on the feedback loop that keeps one X from expressing *Xist* when the other started inactivating; the result would be that one X chromosome is preferentially selected for inactivation as XCI starts in early development. An example of primary skewing is the modulation of XCI initiation by different *X controlling element* (*Xce*) alleles in mice: mouse strains from different genetic backgrounds carry unique *Xce* alleles that result in skewing phenotypes in the progeny of hybrid crosses [139]. The *Xce* segment has not been clearly defined and, so far, the different attempts to map it have located it close or overlapping with the *Xic* [139,140,141,142,143]; reviewed in [144]. One study even suggests that it is not just one locus but that the *Xce* may include different X-linked regions [143]. Importantly, not all those studies use the same approach to measure the “*Xce* effect” and therefore they might be effectively mapping regions that contribute to skewing in XCI patterns for hybrid crosses but not necessarily the *Xce* locus as originally defined [139]. Overall, six competing *Xce* alleles have been proposed, with the order of strength being *a* < *f* < *e* < *b* < *c* < *d*, where *Xce^a^* is the most likely to be inactivated and *Xce^d^* the least likely. A new study has identified an additional allele, reportedly the weakest in the *Xce* allelic series [145]. In *Xce* heterozygotes, the X chromosome carrying the weaker of the two alleles is more likely to be inactivated. The degree of skewing can vary a lot, with cases of mean X inactivation patterns as profound as ∼25:75 in *Xce^c^/Xce^a^* hybrids [140]. Conversely, primary choice is presumably unbiased in *Xce* homozygotes. In humans, a locus homologous to the murine *Xce* has not yet been found, perhaps owing to the unique challenges faced when studying our species [146].

The so-called secondary skewing takes place post-XCI, when cells are selected either for or against depending on which X chromosome they inactivated–reviewed in [147]. This mechanism occurs for instance in individuals that carry X-chromosome-linked variants associated with lethality or restricted survival and is often a hallmark of situations such as being a carrier for X-linked diseases [128,148]. In XX individuals with a structurally abnormal X chromosome (with deletions or duplications) or carrying unbalanced X:autosome translocations, cells that have inactivated the affected chromosome, in such a way that the normal X-chromosome and autosomal dosages are preserved, are positively selected for survival [149]. Another example of this mechanism can be found in XX individuals with Rett syndrome, a neurodevelopmental disorder caused by a mutation in the X-linked gene *MECP2*: unlike XY individuals, who often die of the condition, XX people can survive due to counter-selection of cells carrying the mutated X chromosome [70,150,151]. On the other hand, females with balanced X:autosome rearrangements usually inactivate the normal X chromosome in order to preserve functional expression of autosomal genes on the translocated segment [152]. An example of this situation is the manifestation of clinical symptoms in women heterozygous for mutations in the X-linked *DMD* gene, which, when not functional, results in Duchenne muscular dystrophy, a recessive disease. Several studies have reported cases of women with Xp21; A translocation and preferential inactivation (due to secondary skewing) of the wild-type X chromosome–reviewed in [153]. While inactivation of the wild-type fully-functional copy of the *DMD* gene in these individuals may seem counterintuitive, this mechanism likely prevents monosomy of autosomal genes [154,155], a potentially lethal condition that could occur if the translocated X-chromosome segment (carrying the mutated *DMD* gene) were to be inactivated.

## 7. XCI Choice: The Second Most Important Moment in the Lives of XX Mammals?

Gastrulation has been considered the most important time in our lives, as famously noted by the pioneering developmental biologist Lewis Wolpert, who sadly left us earlier this year [156]. For placental mammals with two X chromosomes, another critical moment in their lives is when, in each of their cells, one of their X chromosomes becomes inactivated. This choice, and the patterns of gene expression that derive from it, can have significant implications. XCI is a major source of diversity within and between XX individuals; it generates stochastic, spatial mosaicism in gene expression across tissues and organs, which can affect their function. While in organs in which many cells perform the same function this might be of little phenotypic consequence, such region-to-region or sometimes left-to-right diversity can lead to uncompensated phenotypes in organs with spatially segregated functions, such as sensory tissues and those part of the central nervous system [132].

In some cases, the choice of which X chromosome to inactivate can make the difference between being healthy and unhealthy, as observed for genetically identical (monozygotic) twins who are phenotypically discordant for genetic diseases carried on the X chromosome. Given that an X chromosome contains 1/20 of the genome’s genes, identical XX twins can potentially differ by up to 5% in the genes they use [157]. For instance, case-studies in XX twins carrying a mutation for Duchenne muscular dystrophy have reported that a skewed XCI pattern renders one twin ill with the disease while the other remains unaffected [158,159,160].

The relationship between XCI choice and the phenotypical manifestation of X-linked mutations could perhaps inform the long-standing question of why (or rather how) rXCI evolved. The process that leads to the choice of which X to inactivate is especially puzzling in mice: why does the system selectively inactivate the Xp in the preimplantation embryo, only to reactivate it and randomly re-inactivate it again upon implantation? An attractive hypothesis is that rXCI could have been selected to limit the consequences of detrimental X-linked mutations in XX individuals; but then why did the same mechanism not evolve in marsupials? Could it be that differences in gene content between the placental and marsupial X are such that marsupial X chromosomes are globally “less affected” by mutations? The answer is probably connected to the different constraints and pressures to which the X chromosome is subject to in these lineages. It will be fascinating to explore these questions in the future.

Finally, is choice reversible? Once established, XCI is mitotically heritable, i.e., daughter cells have the same inactive X as their mother cell, so choice is propagated and skewed spatial patterns maintained. The reactivation of the inactive X chromosome can happen spontaneously (though very rarely), in pathological contexts or in specific developmental moments, such as in the cells that will produce the embryo proper in mice or during oogenesis–reviewed in [125,161,162,163].

Clinically, efforts are ongoing to develop strategies for inducing reactivation of the inactive X in patients–for instance, targeted X-reactivation methods are underway to help young XX individuals affected by Rett syndrome, who carry an active X chromosome harbouring a mutated allele of the *MECP2* gene and an inactive X chromosome harbouring a wildtype allele [151,164,165,166]. “Awakening” their Xi could restore *MECP2* function and cure Rett syndrome. Understanding how choice is established–and how it could be reversed–will therefore remain an important subject of investigation in the X-inactivation field, with wider implications for how we think about epigenetic mechanisms, networks of gene regulation and developmental decisions.

## Figures and Tables

**Figure 1 cells-11-00535-f001:**
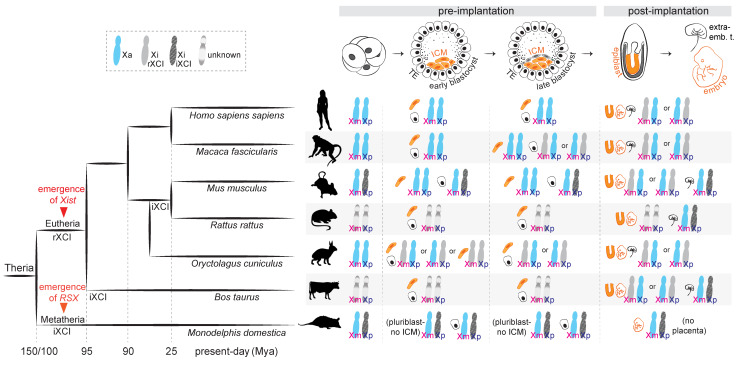
X-chromosome inactivation across species. **Left**: a phylogenetic tree indicating the evolution of random and imprinted XCI and the emergence of long non-coding RNAs *Xist* and *RSX* in Theria. **Right**: X-chromosome inactivation dynamics across development in representative species.

**Figure 2 cells-11-00535-f002:**
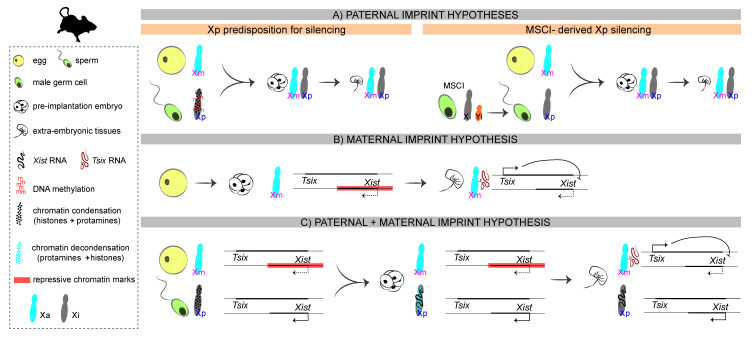
Hypotheses on the molecular nature of the imprint in mice. (**A**) Paternal imprint: The Xp inherits a predisposition for silencing from its life cycle in the male. (**B**) Maternal imprint: In the preimplantation embryo, repressive chromatin marks on the Xm (including the *Xist* promoter region) prevent *Xist* expression on the Xm. In the extra-embryonic tissues of the post-implantation embryo, *Tsix* expression prevents *Xist* upregulation in *cis*. (**C**) Paternal and maternal imprint: A combination of both hypotheses, considering the different chromatin condensation states of the Xp in the sperm and in the paternal pronucleus after fertilization.

**Figure 3 cells-11-00535-f003:**
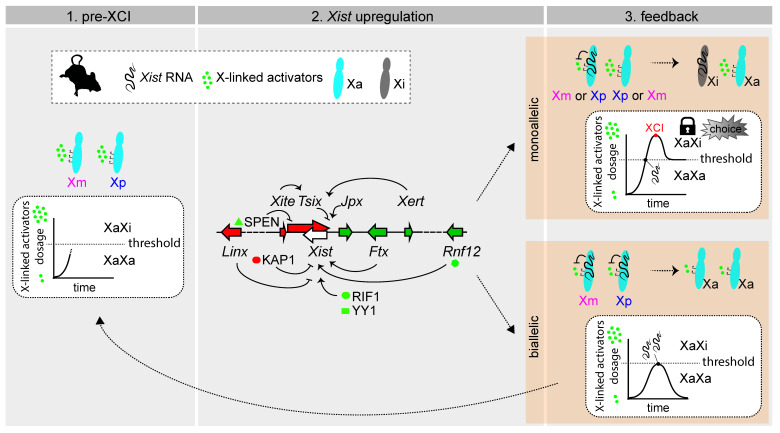
Dynamic model of choice during random XCI. **Left**: Pre-XCI status. Both X chromosomes are active and transcribe X-linked genes. The dose of X-linked activators increases towards the threshold necessary for productive *Xist* upregulation. **Middle**: Biallelic X-chromosome transcription allows the cell to reach the threshold for *Xist* activation. X-linked and autosomal *cis* and *trans* positive and negative regulators influence the initiation of *Xist* upregulation, which can occur on a single X chromosome or on both. Only factors and loci discussed in the text have been included in the figure. **Right (top)**: In cells that have upregulated *Xist* monoallelically, X-wide *cis*-silencing triggered by *Xist* RNA causes a drop in the level of activators, preventing the second chromosome from upregulating *Xist*. Choice is locked-in. Monoallelic *Xist* expression (and *cis*-silencing) has to be sustained, either through enough dosage of activators and/or feedback mechanisms. **Right (bottom)**: In cells that have upregulated *Xist* biallelically, excess *Xist* expression triggers rapid downregulation of X-linked activators on both X chromosomes, and this drop in levels below the threshold causes *Xist* expression to switch off. Both X chromosomes remain active and the process must start again.

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
