# Peer review of "Mechanisms of Choice in X-Chromosome Inactivation"

_cells, 2022, doi:10.3390/cells11030535_

Round 1

Reviewer 1 Report

I really enjoyed reading this review and I think it can essentially be published as is. I have a few small suggestions.

- Icons and text in the figures are in general a bit small and difficult to read.

- Fig. 1 is not easy to understand. I could maybe help mark random and imprinted XCI with two different colors, e.g. surround the chromosomes with colored boxes or underscores?

- The section numbering is not correct.

- For biallelic Xist upregulation and XCI our study from last year (Pacini et al, Nat Comm, 2021) might be a better reference than Mutzel et al, 2019, since it characterizes Xist levels and XCI in detail by allele-specific scRNA-seq.

- Our preprint Gjaltema et al, just appeared in a peer-reviewed form in Molecular Cell.

- The wrong paper is cited in line 523: Should be De mello, et al, 2017, Scientific reports

Reviewer 2 Report

Furlan and Galupa present a very well written and comprehensive review of our current understanding of X-chromosome inactivation with a specific focus on how Xs are chosen.  They do a particularly good job of pointing out when potential mechanisms have been debated in the literature, which is essential for a review article.   Although not very intuitive, the figures present a large amount of information in a reasonable way. I have no major critiques (see minor comments below) and I believe this review will add significant value to the field.  The one modestly significant critique is the last line of the abstract stating “We also call for a revised manner in which to think about choice…” – this is NOT effectively done in the manuscript. What is the new idea they present?  This is not needed for the review ms, so this line should either be removed, or they should add a paragraph at the end stating what their new idea is.

Minor edits/critiques:

  1. Line 46 – redundant use of “different”
  2. Line 54: rearrange sentence to read “While some species…the same X, other species choose an X randomly.”
  3. Line 167-169: Try to rephrase for clarity
  4. Line 195 change to “…. For the imprint to be functional.”
  5. Lines 519 – 523: This sentence is confusing because it makes it seem like something published in 2010 was a reanalysis of a dataset published in 2016 – please clarify.  
  6. Line 582 – may --> many  
  7. Line 635 change “what” to “which”
  8. Line 649/650 move “not” to read “…but then why did the same mechanism NOT evolve in marsupials?”

8 Figure 2: repressive chromatin marks are hard to distinguish  

Reviewer 3 Report

This is a well-written, informative piece.  I have no reservations about publishing as is.

Author Response

Thank you very much for your revisions!